# HYPERBOLIC GEOMETRY OF REASONING: PROBING LLM HIDDEN STATES

**Arnav Raj**
Department of Computer Science and Engineering
Indian Institute of Technology Delhi
New Delhi, India
`arnav.raj.cs522@cse.iitd.ac.in`

## ABSTRACT

Large language models with chain-of-thought reasoning exhibit hierarchical dependencies, yet the geometric structure of these representations remains underexplored. We probe DeepSeek-R1 (reasoning-specialized) and Qwen2.5 (standard instruction-tuned) on PrOntoQA logical reasoning tasks, comparing Euclidean and hyperbolic probe geometries. Hyperbolic probes maintain robust performance across all layers, while Euclidean probes exhibit late-layer degradation specific to reasoning models, stable at early layers but degrading substantially at the final layer. Standard instruction-tuned models show no such degradation. We further show that probing "thinking tokens" (reasoning-critical tokens identified via linguistic markers) concentrates hierarchical information far more effectively than uniform pooling at the compressed final layer. Layer-wise activation statistics provide statistical evidence linking representational compression to the geometry-dependent performance gap. These findings suggest that hyperbolic geometry provides important robustness advantages for probing reasoning representations, conditional on model architecture.

## 1 INTRODUCTION

Chain-of-thought (CoT) reasoning (Wei et al., 2022) in LLMs generates hierarchical dependencies where premises build upon one another to form conclusions. Understanding the geometric structure of these representations is fundamental for interpretability and reliability. Hyperbolic geometry naturally accommodates tree-like structures through exponential volume growth (Nickel & Kiela, 2017; Ganea et al., 2018; Sarkar, 2011), and recent empirical analysis suggests LLM embeddings exhibit intrinsic tree-like ($\delta$-hyperbolic) structure (He et al., 2025b), prompting the question: *Do hyperbolic probes better capture hierarchical reasoning than Euclidean alternatives?*

Prior work developed structural probes recovering syntax trees from Euclidean distances (Hewitt & Manning, 2019) and extended probing to hyperbolic spaces for BERT (Chen et al., 2021), while others probed factual truth (Marks & Tegmark, 2024; Burns et al., 2023) or analyzed reasoning dynamics without explicit geometric frameworks (Qian et al., 2025). However, to the best of our knowledge, hyperbolic probes have not yet been applied to *reasoning-specialized* decoder LLMs, which may encode hierarchies fundamentally differently than standard instruction-tuned models. Furthermore, recent work identified "thinking tokens" as sparse carriers of reasoning dynamics (Qian et al., 2025), but their *geometric* role—whether they concentrate hierarchical structure in hidden states—remains unexplored. Beyond immediate interpretability, understanding geometric encoding of hierarchical computation advances the broader mechanistic interpretability agenda (Elhage et al., 2022), potentially informing circuit-level analysis of reasoning. These gaps motivate this preliminary cross-model investigation.

**Contributions.** (1) We demonstrate that hyperbolic probes achieve consistently high performance across models and layers, while Euclidean probes exhibit model-dependent late-layer degradation. (2) We identify final-layer Euclidean degradation as specific to reasoning-specialized models; standard models maintain robust performance. (3) We validate that "thinking tokens"—identified via reasoning markers following Qian et al. (2025)—concentrate hierarchical information at the final

layer, outperforming uniform pooling. (4) We provide statistical evidence linking representational compression at the final layer to the observed geometry-dependent performance gap.

## 2 PRELIMINARIES

**Why Hyperbolic Geometry for Hierarchies.** In Euclidean space $\mathbb{R}^d$, the volume of a ball grows polynomially as $r^d$. In hyperbolic space of curvature $-c$, volume grows exponentially as $e^{(d-1)\sqrt{c}\,r}$. This mirrors the structure of trees, where the number of nodes grows exponentially with depth. Sarkar (2011) proved that any finite tree embeds in the 2D hyperbolic plane with arbitrarily low distortion—impossible in any fixed-dimensional Euclidean space. Empirically, He et al. (2025b) measured $\delta$-hyperbolicity in LLM embedding spaces ($\delta \in [0.07, 0.20]$ across models), suggesting these representations possess intrinsic non-Euclidean structure amenable to hyperbolic analysis.

**The Poincaré Ball Model.** We work in the Poincaré ball $\mathbb{B}_c^d = \{x \in \mathbb{R}^d : c\|x\|^2 < 1\}$, a model of hyperbolic space parameterized by curvature $c > 0$ (corresponding to sectional curvature $-c$). A key property is that distances grow rapidly near the boundary: points close in Euclidean norm can be far apart in hyperbolic distance. This naturally encodes depth—shallow nodes map near the origin where space is plentiful, while deeper nodes map toward the boundary where exponential volume accommodates exponential branching.

**Structural Probes.** Hewitt & Manning (2019) introduced structural probes: learned linear transformations of hidden states such that pairwise Euclidean distances in the transformed space approximate tree distances. Chen et al. (2021) extended this to the Poincaré ball, replacing Euclidean with hyperbolic distance, and showed improved recovery of dependency hierarchies in BERT. We extend this framework to reasoning-specialized decoder LLMs, where the "tree" structure arises from chains of logical deduction rather than syntactic dependencies.

## 3 METHOD

**Hyperbolic Probe Architecture.** Given layer activations $h \in \mathbb{R}^{3584}$, we map to the Poincaré ball $\mathbb{B}_c^d$ ($d = 5$, $c = 0.5$). First, we normalize and project: $h' = \text{LayerNorm}(h)$, $z_{\text{euc}} = \mathbf{W}h' + b$ with spectral normalization (Miyato et al., 2018). Then we apply bounded scaling: $z_{\text{scaled}} = \sigma(\alpha)\tanh(z_{\text{euc}})$ where sigmoid $\sigma(\alpha)$ is learnable (init 0.95). Finally, we map to hyperbolic space: $z_{\text{hyp}} = \exp_0^c(z_{\text{scaled}})$ where

$$\exp_0^c(v) = \tanh\left(\frac{\sqrt{c}}{2}\|v\|\right)\frac{v}{\sqrt{c}\|v\|} \tag{1}$$

**Numerical Stability.** We apply Maximum Distance Rescaling (MDR) (Bdeir et al., 2025) ($r_{\max} = 15$) before the exponential map to prevent gradient explosion. Curvature $c = 0.5$ balances structure and stability. The Poincaré distance is $d_{\mathbb{P}}(z_i, z_j) = \frac{1}{\sqrt{c}}\text{arcosh}(1 + 2c\frac{\|z_i - z_j\|^2}{(1-c\|z_i\|^2)(1-c\|z_j\|^2)})$.

**Training Objective.** We minimize stress-normalized loss: $\mathcal{L} = \sum_{i \neq j}(d_{\mathbb{P}}(z_i, z_j) - d_{\text{graph}}(i,j))^2 / \sum_{i \neq j} d_{\text{graph}}(i,j)^2$, where $d_{\text{graph}}(i,j) = |\text{depth}_i - \text{depth}_j|$ treats reasoning depth as a 1D ordinal variable. **Euclidean baseline:** L2 distance $\|z_i - z_j\|_2$ on the same projected embeddings. Training: 100 epochs, Adam lr=$10^{-3}$, batch=64, 5-fold CV.

**Dataset. PrOntoQA** (Saparov & He, 2023): 1000 logical reasoning problems with depths 1–5, forming linear reasoning chains. Reasoning depth serves as the ground-truth distance (defined above). Templated language minimizes linguistic confounds.

**Experimental Design. Models:** DeepSeek-R1-Distill-Qwen-7B (DeepSeek-AI, 2025) (reasoning-specialized with CoT training) vs. Qwen2.5-7B-Instruct (Qwen Team, 2024) (standard instruction-tuned). We analyze 8 layers (L8, L12, L16, L19, L21, L23, L25, L27), spanning early, middle, and late processing with denser coverage near the final layer (full results in Appendix B). **Metric:** Spearman $\rho$ between predicted and ground-truth distances. All experiments use fixed seed 42.

Table 1: Final layer (L27) probe performance on PrOntoQA (5-fold CV, all cross-validation std $< 0.01$). Distortion = mean absolute distance error; lower is better.

| Model | Spearman $\rho \uparrow$ | | Distortion $\downarrow$ | |
|---|---|---|---|---|
| | Euc | Hyp | Euc | Hyp |
| DeepSeek-R1 (reasoning) | 0.488 | **0.967** | 0.562 | **0.090** |
| Qwen2.5 (standard) | 0.955 | **0.967** | 0.139 | **0.104** |

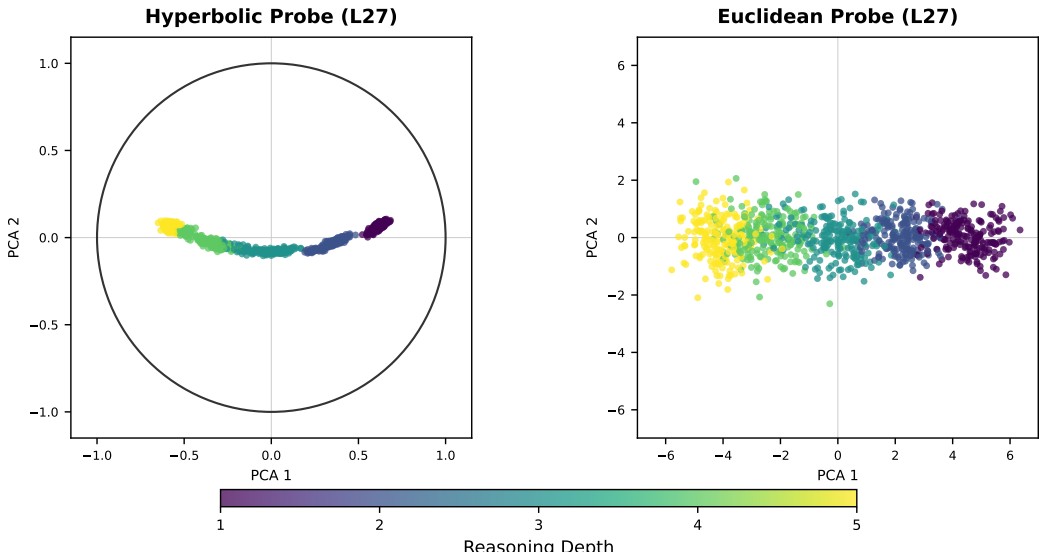

Figure 1: Layer 27 probe embeddings (DeepSeek, PrOntoQA) projected to 2D via PCA, colored by reasoning depth. The hyperbolic probe (left) preserves depth ordering as a smooth gradient within the Poincaré disk, while the Euclidean probe (right) scatters depth classes, illustrating the $6\times$ distortion gap in Table 1.

## 4    RESULTS

### 4.1    CROSS-MODEL VALIDATION

Table 1 presents final layer performance. Hyperbolic probes achieve consistently high performance ($\rho \approx 0.97$) across both models. However, Euclidean probes exhibit substantial degradation in DeepSeek on PrOntoQA ($\rho = 0.488$), while Qwen maintains robust performance ($\rho = 0.955$). Distortion—mean absolute distance error—reveals an even sharper contrast: DeepSeek Euclidean distortion (0.562) is $6\times$ higher than hyperbolic (0.090), while Qwen shows comparable distortion across geometries. Since the 1D ordinal target metric embeds isometrically in both geometries, this advantage must arise from the *representation* structure itself—the model's internal geometry genuinely favors hyperbolic decoding. This pattern suggests that late-layer Euclidean degradation is specific to reasoning-specialized architectures. Figure 1 visualizes this contrast.

**Progressive Degradation Pattern.** Complete layer-wise analysis (Appendix B) reveals systematic Euclidean degradation in DeepSeek: stable performance at early/mid layers (L8–L21: $\rho \approx 0.97$), initial degradation at Layer 23 ($\rho = 0.84$), partial recovery at Layer 25 ($\rho = 0.91$), and substantial degradation at the final layer ($\rho = 0.49$). Hyperbolic performance remains robust throughout ($\rho \geq 0.90$ across all layers). With only 5 unique distance values ($|d_i - d_j| \in \{0, \ldots, 4\}$), rank correlation $\rho$ saturates near 1.0 given only 5 ordinal distance levels. Distortion, which measures absolute distance preservation, reveals richer layer-dependent variation: Euclidean distortion increases $19\times$ from L8 to L27 while hyperbolic remains stable (Figure 2, Appendix). Crucially, Qwen Euclidean shows no degradation at any layer ($\rho \approx 0.96$ across all layers).

Table 2: Layer 27 token selection ablation (DeepSeek-R1 + PrOntoQA, hyperbolic probe). Thinking tokens concentrate hierarchical information at the final layer.

| Selection Method | Spearman $\rho$ | vs. All-Pool |
|---|---|---|
| Thinking Tokens | **0.871** | +124% |
| Last Token | 0.468 | +20% |
| All-Token Pool | 0.390 | baseline |

**Statistical Evidence: Representational Compression.** To characterize the hypothesized late-layer compression, we computed layer-wise statistics on DeepSeek activations (Table 5). Activation norms increase monotonically from L8 to L25, then *decrease sharply* at the final layer ($-41\%$). Simultaneously, norm variance explodes ($+214\%$), participation ratio—a measure of effective dimensionality—drops from 45.5 to 25.8 ($-43\%$), and isotropy increases $\sim 20\times$. Together, these indicate severe representational compression: reduced effective dimensionality and loss of directional diversity at the output layer. Hyperbolic geometry, with its exponential volume growth, accommodates this compressed structure where Euclidean distances fail. Crucially, Qwen also compresses at L27 but far less severely (participation ratio $-29\%$ vs. $-43\%$; Appendix B.2), explaining why its Euclidean probes remain robust.

### 4.2 THINKING TOKEN SELECTION

Table 2 presents Layer 27 performance for different token selection strategies on DeepSeek with PrOntoQA. Probing thinking tokens—identified via reasoning marker patterns ("Wait", "So", "Therefore") following Qian et al. (2025)—yields $\rho = 0.871$, substantially outperforming last-token pooling ($\rho = 0.468$) and uniform all-token pooling ($\rho = 0.390$). These results provide geometric validation of the information-theoretic findings that reasoning dynamics concentrate in sparse, identifiable tokens constituting merely 0.5–5% of the sequence (Qian et al., 2025) (our pattern-based approach yields 6.7%; see Appendix A.4).

**Cross-Layer Token Selection.** We additionally analyzed thinking token selection across layers (Appendix B.1). The thinking token advantage is concentrated at the final layer ($\Delta\rho = +0.481$ at L27), with moderate benefit at L21 ($\Delta\rho = +0.224$). At intermediate layers (L19, L23, L25), thinking tokens perform worse than uniform pooling (negative $\Delta\rho$), possibly because reasoning markers are less informative at layers where representations are not yet compressed. This pattern suggests that hierarchical information concentrates in thinking tokens specifically at the representationally compressed final layer (Appendix A.4).

## 5 DISCUSSION AND RELATED WORK

**Hyperbolic Representation Learning.** Nickel & Kiela (2017) established that 5D Poincaré embeddings match 200D Euclidean performance on hierarchical data; their follow-up Lorentz model (Nickel & Kiela, 2018) improved numerical stability. Ganea et al. (2018) derived hyperbolic versions of core neural network operations, enabling end-to-end hyperbolic learning. Subsequent work extended to vision (Bdeir et al., 2024) with stable training methods (Bdeir et al., 2025). He et al. (2025b) empirically measured $\delta$-hyperbolicity ($\delta \in [0.07, 0.20]$) in LLM embeddings. Our work applies these frameworks to probe reasoning structure in hidden states.

**Probing Internal Representations.** Hewitt & Manning (2019) introduced structural probes recovering syntax trees from Euclidean distances; Chen et al. (2021) extended this to Poincaré space, showing hyperbolic probes better recover dependency hierarchies in BERT. Marks & Tegmark (2024) identified Euclidean truth directions for factual statements. Chen et al. (2024) employed eigenvalue analysis on mid-layer representations for hallucination detection, and Chain-of-Embedding (Wang et al., 2025) treats hidden states as a "latent thinking path" for self-evaluation. Concurrent work by Zhong et al. (2026) extends probing from chains to DAG structures. Our work differs by: (1) applying hyperbolic probes to *reasoning-specialized* decoder language models (extending Chen et al. (2021) beyond BERT), (2) comparing reasoning-specialized vs. standard models, and (3) identifying geometry-dependent degradation at the final layer.

**Reasoning Dynamics.** Qian et al. (2025) identified thinking tokens via mutual information analysis, showing they constitute merely 0.5–5% of generated tokens yet concentrate reasoning dynamics. Zhang et al. (2025) demonstrated linear separability of correctness in reasoning chunks (AUROC $> 0.85$), and Afzal et al. (2025) showed hidden states encode CoT success before completion. Our geometric analysis provides complementary validation: thinking tokens not only carry high mutual information but also concentrate geometric hierarchical structure.

**Geometric Structure at Scale.** A natural question is whether geometric inductive biases retain value in billion-parameter models (He et al., 2025a). Our findings suggest a nuanced answer: hyperbolic geometry offers clear robustness advantages for reasoning-specialized models, while Euclidean representations suffice for standard instruction-tuned architectures. This conditional advantage aligns with recent work showing hyperbolic fine-tuning improves complex reasoning (Yang et al., 2025), suggesting geometric structure may become increasingly relevant as models specialize for hierarchical tasks.

**Connection to Anisotropy and Compression.** Ethayarajh (2019) first documented anisotropy in contextual representations, and Razzhigaev et al. (2024) showed bell-shaped anisotropy across decoder layers. Barbero et al. (2024) showed decoder transformers suffer information over-squashing where distinct inputs yield arbitrarily close final-token representations, and Sun et al. (2025) documented that deeper layers in modern LLMs are less effective due to output variance explosion. Our findings align with these observations, yet our cross-model results show the degradation is not inherent (Machina & Mercer, 2024) but conditioned on reasoning-specific optimization.

**Limitations.** As a preliminary investigation, this work has several limitations: (1) Both models share the Qwen2.5 backbone, so our cross-model comparison reflects fine-tuning regime differences rather than architectural differences. (2) We evaluate only 7B-parameter models; scaling to 70B+ may reveal different patterns. (3) PrOntoQA provides controlled 1D chain structure; extending to branching hierarchies and real-world reasoning tasks is an important next step. (4) We have not explored cross-domain generalization. (5) While layer statistics provide evidence for representational compression, full mechanistic understanding requires circuit-level analysis. (6) Models were loaded with 4-bit quantization, which may affect activation distributions.

## 6 CONCLUSION

We investigated whether hyperbolic geometry more effectively captures hierarchical reasoning structure in LLM hidden states. Through cross-model experiments on PrOntoQA, we found that hyperbolic probes maintain robust performance across all conditions, while Euclidean probes exhibit substantial late-layer degradation specific to reasoning-specialized models. We further showed that thinking tokens concentrate hierarchical information at the compressed final layer ($\rho = 0.871$ vs. 0.390 for uniform pooling), and provided statistical evidence linking representational compression to this geometry-dependent gap. Our findings suggest that geometric structure plays a conditional yet important role: hyperbolic probes offer clear advantages precisely where reasoning-specialized models develop compressed representations. Future research should explore larger models and extension to branching hierarchies (Zhong et al., 2026).

### REPRODUCIBILITY STATEMENT

Code and datasets available at https://github.com/deadsmash07/hyperbolic-reasoning-probe. All experiments employ fixed seed 42. Hyperparameters: $d = 5$, $c = 0.5$ (ablations in Appendix C); full training configuration in Appendix A.

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

# A EXPERIMENTAL DETAILS

## A.1 HARDWARE AND SOFTWARE ENVIRONMENT

All experiments conducted on NVIDIA H100 GPUs (80GB HBM3) with CUDA 12.1. Software stack: PyTorch 2.1.0, geoopt 0.5.0 (hyperbolic operations), transformers 4.36.0 (model loading), bitsandbytes 0.41.0 (quantization). Models loaded via HuggingFace Hub with 4-bit quantization to fit 7B models on single GPU. Activation extraction: batch size 1 for memory consistency across layers.

## A.2 MODEL SPECIFICATIONS

**DeepSeek-R1-Distill-Qwen-7B:** 28 transformer layers, hidden dimension 3584, 28 attention heads. Distilled from DeepSeek-R1 (671B parameters) via supervised fine-tuning on chain-of-thought traces. Generates explicit reasoning steps via special "$\langle$think$\rangle$" tokens. **Qwen2.5-7B-Instruct:** 28 layers, hidden dim 3584, 28 heads. Instruction-tuned without specialized reasoning training. Pre-trained on general web corpus; fine-tuned with instruction-following data.

## A.3 DATASET GENERATION PROTOCOLS

**PrOntoQA Generation:** Created 1000 samples with depths uniformly distributed across $\{1, 2, 3, 4, 5\}$ (200 per depth). Template structure: Facts: "All $X$ are $Y$" (depth times), Query: "$Z$ is $X_1$. Is $Z$ a $Y_{\text{final}}$?" Vocabulary: 50 unique predicates, 100 unique entities, ensuring no vocabulary overlap between generated samples. Ground-truth distance = reasoning chain depth. Balanced TRUE/FALSE labels (50%/50%).

## A.4 THINKING TOKEN IDENTIFICATION

Following the conceptual framework of Qian et al. (2025), we identify thinking tokens via regex patterns matching reasoning markers: `"Wait|Hmm|Let me|So|Therefore|Thus|Hence|Because|Since"`. This yields an average of 20.7 tokens per sample (6.7% of sequence length). While Qian et al. (2025) use mutual information peaks, our pattern-based approach provides a practical approximation explicitly targeting reasoning steps.

## A.5 HYPERPARAMETER SELECTION RATIONALE

**Embedding Dimension ($d$):** Tested $d \in \{2, 4, 5, 8, 16, 32\}$ on DeepSeek + PrOntoQA L27. Hyperbolic results: $d = 2$ ($\rho = 0.936$), $d = 5$ ($\rho = 0.966$) optimal, $d \geq 16$ ($\rho \approx 0.97$) diminishing returns. Selected $d = 5$ for optimal balance of performance and computational cost.

**Curvature ($c$):** Tested $c \in \{0.1, 0.3, 0.5, 0.7, 1.0\}$. Lower curvature ($c = 0.1$, $\rho = 0.950$) approaches Euclidean behavior. Optimal at $c = 0.5$ ($\rho = 0.966$). Higher curvature ($c = 1.0$, $\rho = 0.960$) shows slight degradation due to points concentrating near the Poincaré ball boundary.

**Spectral Normalization:** Applied to projection matrix $\mathbf{W}$ via power iteration (5 iterations per forward pass). Normalizes largest singular value to 1.0, preventing gradient explosion common in hyperbolic training (Bdeir et al., 2025).

**Maximum Distance Rescaling (MDR):** Following Bdeir et al. (2025), apply $\boldsymbol{v}_{\text{rescaled}} = \boldsymbol{v} \cdot \frac{\tanh(\|\boldsymbol{v}\|/15) \cdot 15}{\|\boldsymbol{v}\|}$ to prevent numerical overflow in exponential map when $\|\boldsymbol{v}\| \gg 1$.

## A.6 TRAINING PROCEDURES

**Cross-Validation:** 5-fold stratified CV with 80/20 train/test splits. Stratification ensures balanced depth distribution per fold. Each fold trained independently with separate random initialization.

**Optimization:** Adam optimizer with $\beta_1 = 0.9$, $\beta_2 = 0.999$, $\epsilon = 10^{-8}$. Learning rate $10^{-3}$ (constant, no decay). Batch size 64. Gradient clipping: max global norm 1.0. Early stopping: patience=10 epochs on validation MSE (min delta=$10^{-4}$).

**Loss Function:** Stress-normalized loss (Kruskal stress, a standard multidimensional scaling metric):

$$\mathcal{L} = \frac{\sum_{i \neq j} \left(d_{\text{pred}}(i,j) - d_{\text{true}}(i,j)\right)^2}{\sum_{i \neq j} d_{\text{true}}(i,j)^2}$$

where $d_{\text{pred}}$ denotes $d_{\mathbb{P}}$ (hyperbolic) or $\|\cdot\|_2$ (Euclidean) and $d_{\text{true}} = d_{\text{graph}}$.

## B    COMPLETE LAYER-WISE ANALYSIS

Table 3 presents complete layer-wise results for all experimental conditions across both geometries. Figure 2 visualizes the distortion trends.

Table 3: Complete layer-wise Spearman $\rho$ and distortion for PrOntoQA. While hyperbolic $\rho$ remains near-perfect across all layers, distortion reveals progressive Euclidean degradation in DeepSeek. Notably, Euclidean distortion is *lower* than hyperbolic at early layers, confirming both probes are layer-sensitive.

| | DeepSeek-R1 | | | | Qwen2.5 | | | |
| | Spearman $\rho \uparrow$ | | Distortion $\downarrow$ | | Spearman $\rho \uparrow$ | | Distortion $\downarrow$ | |
| **Layer** | Euc | Hyp | Euc | Hyp | Euc | Hyp | Euc | Hyp |
|---|---|---|---|---|---|---|---|---|
| 8  | 0.970 | 0.967 | 0.029 | 0.112 | 0.970 | 0.967 | 0.028 | 0.100 |
| 12 | 0.970 | 0.967 | 0.034 | 0.114 | 0.960 | 0.967 | 0.147 | 0.115 |
| 16 | 0.970 | 0.967 | 0.058 | 0.117 | 0.961 | 0.895 | 0.124 | 0.308 |
| 19 | 0.970 | 0.967 | 0.072 | 0.109 | 0.970 | 0.911 | 0.027 | 0.293 |
| 21 | 0.970 | 0.908 | 0.064 | 0.285 | 0.970 | 0.967 | 0.043 | 0.105 |
| 23 | 0.842 | 0.967 | 0.521 | 0.112 | 0.970 | 0.967 | 0.050 | 0.112 |
| 25 | 0.906 | 0.967 | 0.209 | 0.101 | 0.970 | 0.966 | 0.066 | 0.144 |
| 27 | **0.488** | **0.967** | **0.562** | **0.090** | 0.955 | 0.967 | 0.139 | 0.104 |

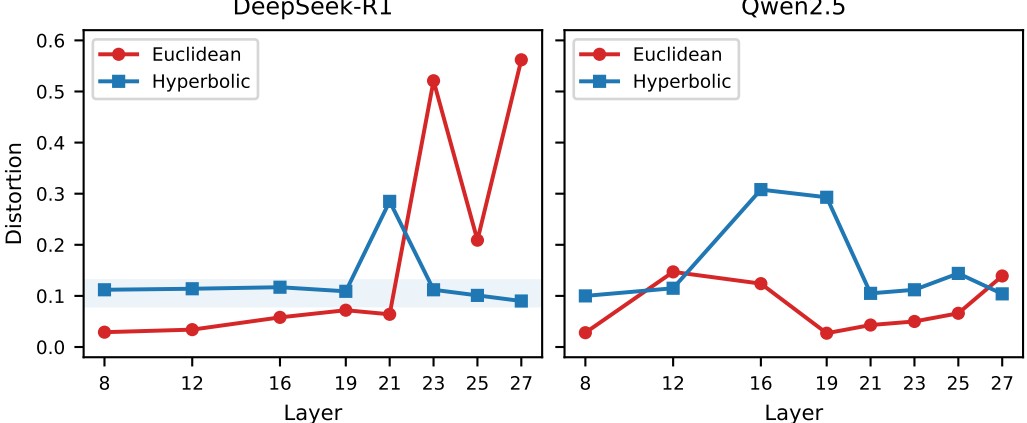

Figure 2: Layer-wise distortion for Euclidean and hyperbolic probes. DeepSeek Euclidean distortion diverges sharply at late layers while hyperbolic remains bounded (0.09–0.12). Qwen shows no Euclidean divergence. At early layers, Euclidean distortion is lower than hyperbolic.

**Degradation Pattern Analysis:** The three-phase Euclidean degradation pattern described in Section 4.1 is quantified at full precision in Table 3. Distortion corroborates with finer granularity: Euclidean distortion increases progressively from 0.029 (L8) through 0.072 (L19) to 0.562 (L27), whereas hyperbolic distortion remains in the range 0.090–0.117 across all non-anomalous layers.

**Anomalous Hyperbolic Behavior:** The hyperbolic probe shows an isolated dip at L21 ($\rho = 0.908$, distortion 0.285) before recovering fully at L23–L27. This may reflect a transitional layer where

hierarchical information is reorganized, temporarily disrupting geometric structure. Further mechanistic investigation is needed.

**Qwen Hyperbolic Dips:** Qwen's hyperbolic probe shows performance dips at L16 ($\rho = 0.895$) and L19 ($\rho = 0.911$), both recovering fully by L21 ($\rho = 0.967$). These mid-layer dips, absent in DeepSeek, may reflect differences in how standard instruction-tuned models organize hierarchical information at intermediate processing stages. Notably, these dips do not persist to the final layer, and Qwen's Euclidean probe remains stable throughout, suggesting the dips represent transient geometric reorganization rather than systematic failure of hyperbolic probing.

## B.1 Cross-Layer Token Selection Analysis

Table 4 shows thinking token advantages across all late layers for DeepSeek + PrOntoQA.

Table 4: Token selection across layers (DeepSeek + PrOntoQA, hyperbolic probe). Thinking token advantage is concentrated at the final layer (L27).

| Layer | Thinking | Last Token | All-Pool | $\Delta$ (Think–All) |
|---|---|---|---|---|
| 19 | 0.153 | 0.808 | 0.402 | $-0.249$ |
| 21 | 0.702 | 0.681 | 0.479 | $+0.224$ |
| 23 | 0.263 | 0.701 | 0.363 | $-0.100$ |
| 25 | 0.228 | 0.581 | 0.446 | $-0.218$ |
| 27 | **0.871** | 0.468 | 0.390 | **+0.481** |

**Layer-Specific Concentration:** The pattern in Table 4 corroborates Section 4.2: the thinking token advantage emerges specifically under representational compression (L27) or at transitional layers (L21), but is absent at intermediate layers where representations are less compressed.

## B.2 Layer-wise Statistics

Table 5 presents layer-wise activation statistics for DeepSeek on PrOntoQA, providing statistical evidence for the hypothesized representational compression.

Table 5: Layer-wise activation statistics (DeepSeek + PrOntoQA). Participation ratio = $(\sum \lambda_i)^2 / \sum \lambda_i^2$ measures effective dimensionality; isotropy = $\lambda_{\min}/\lambda_{\max}$ measures directional uniformity. The final layer shows sharp representational compression: norm drops $41\%$, variance increases $214\%$, and effective dimensionality decreases $43\%$.

| Layer | Mean Norm | Std Norm | Part. Ratio | Isotropy |
|---|---|---|---|---|
| 8 | 889 | 8.5 | 36.6 | 0.0004 |
| 16 | 987 | 17.1 | 40.8 | 0.0008 |
| 21 | 1059 | 21.9 | 43.4 | 0.0018 |
| 25 | 1333 | 39.7 | 45.5 | 0.0049 |
| **27** | **782** | **124.4** | **25.8** | **0.096** |

**Interpretation:** These statistics corroborate the compression hypothesis from Section 4.1: the final layer exhibits simultaneous norm decrease, variance explosion, dimensionality collapse, and isotropy increase—all consistent with representational compression for output generation.

**Cross-Model Comparison.** We also computed layer-wise statistics for Qwen (Table 6). Both models exhibit final-layer compression, but DeepSeek's is far more severe: norm drops $41\%$ vs. $19\%$, variance increases $214\%$ vs. $91\%$, and participation ratio decreases $43\%$ vs. $29\%$. Notably, even Qwen's most compressed layer (PR = 43.1 at L27) retains $67\%$ more effective dimensionality than DeepSeek's (PR = 25.8), suggesting Qwen preserves sufficient geometric structure for Euclidean probes to operate effectively. This difference in compression severity provides a quantitative explanation for the model-specific Euclidean degradation.

Table 6: Layer-wise activation statistics (Qwen + PrOntoQA). Qwen exhibits final-layer compression but less severely than DeepSeek (Table 5): participation ratio drops $29\%$ vs. $43\%$.

| Layer | Mean Norm | Std Norm | Part. Ratio | Isotropy |
|---|---|---|---|---|
| 8 | 203 | 1.0 | 42.9 | 0.001 |
| 16 | 223 | 4.5 | 53.8 | 0.002 |
| 21 | 239 | 7.8 | 49.8 | 0.005 |
| 25 | 324 | 15.8 | 60.5 | 0.012 |
| **27** | **263** | **30.2** | **43.1** | **0.116** |

## C  ABLATION STUDIES

### C.1  COMPREHENSIVE DIMENSION ABLATION

Tested $d \in \{2, 4, 5, 8, 16, 32\}$ on DeepSeek + PrOntoQA L27, comparing Euclidean and hyperbolic (best curvature per dimension):

Table 7: Dimension ablation on DeepSeek + PrOntoQA L27. Hyperbolic probes are dimension-efficient ($d$=2: $\rho$=0.936), while Euclidean performance degrades at higher dimensions.

| Dim | Euc $\rho$ | Hyp $\rho$ | Best $c$ | Hyp Dist. |
|---|---|---|---|---|
| 2 | 0.750 | 0.936 | 1.0 | 0.210 |
| 4 | 0.767 | 0.954 | 0.5 | 0.179 |
| 5 | 0.903 | **0.966** | 0.5 | 0.156 |
| 8 | 0.826 | 0.967 | 0.5 | 0.148 |
| 16 | 0.363 | 0.970 | 0.5 | 0.143 |
| 32 | 0.207 | 0.970 | 0.5 | 0.136 |

**Key findings:** (1) Hyperbolic probes are dimension-efficient: $d = 2$ already achieves $\rho = 0.936$, while Euclidean requires $d = 5$ to reach $\rho = 0.903$. (2) Euclidean performance *degrades* at higher dimensions ($d = 16$: 0.363, $d = 32$: 0.207), possibly due to increased sensitivity to the compressed L27 representations in higher-dimensional spaces. (3) We select $d = 5$, $c = 0.5$ as detailed in §A.

### C.2  CURVATURE ABLATION

Tested $c \in \{0.1, 0.3, 0.5, 0.7, 1.0\}$ on DeepSeek + PrOntoQA L27 ($d = 5$):

Table 8: Curvature ablation on DeepSeek + PrOntoQA L27 ($d$=5). Moderate curvature ($c$=0.5) provides optimal balance between hyperbolic structure and numerical stability.

| Curvature | $\rho$ | Distortion |
|---|---|---|
| 0.1 | 0.950 | 0.187 |
| 0.3 | 0.963 | 0.162 |
| 0.5 | **0.966** | **0.156** |
| 0.7 | 0.965 | 0.159 |
| 1.0 | 0.960 | 0.168 |

**Analysis:** The curvature–performance relationship reveals an interpretable pattern: at $c = 0.1$, the Poincaré ball approximates Euclidean space and performance suffers; at $c = 1.0$, embeddings concentrate near the ball boundary, reducing effective resolution. The optimum at $c = 0.5$ balances geometric expressiveness with numerical stability.

## D  FUTURE DIRECTIONS

**Scale.** Our analysis covers 7B-parameter models; extending to 70B+ models may reveal whether late-layer degradation intensifies or attenuates with scale. Of particular interest is whether larger

reasoning models (e.g., DeepSeek-R1 at 671B) exhibit similar geometric signatures, or whether increased capacity alleviates the representational compression we observe.

**Complex Hierarchies.** PrOntoQA provides linear 1D chains. Real-world reasoning often involves branching DAG structures (Zhong et al., 2026), requiring methodological extensions beyond pairwise distance probing. Tree-structured datasets with known branching factors would enable direct evaluation of whether hyperbolic embeddings recover branching topology, not merely depth ordering.

**Geometric Fine-tuning.** Our findings are diagnostic: we identify where geometric mismatches arise, but do not intervene. A natural extension is hyperbolic-aware training objectives. Recent work on hyperbolic fine-tuning (Yang et al., 2025) demonstrates improved reasoning performance; combining such approaches with our probing framework could enable both diagnosis and targeted correction of geometric bottlenecks.

**Alternative Geometric Models.** We use the Poincaré ball, but the Lorentz model (Nickel & Kiela, 2018) offers superior numerical stability for high-curvature regimes. Mixed-curvature spaces that adapt geometry per layer may better capture the heterogeneous structure we observe across the network—early layers favoring Euclidean geometry while late layers require hyperbolic.

**Mechanistic Understanding.** While layer statistics provide evidence for representational compression, full mechanistic understanding requires circuit-level analysis identifying specific attention heads and MLPs responsible for the observed compression, and explaining why reasoning-specialized fine-tuning amplifies it.

**Cross-Domain Transfer.** Testing whether probes trained on logical reasoning transfer to mathematical or commonsense reasoning would establish generality beyond the controlled PrOntoQA setting.

