# OpenReview forum: "Hyperbolic Geometry of Reasoning: Probing LLM Hidden States"
_ICLR.cc/2026/Workshop/GRaM — ICLR 2026 Workshop GRaM Poster_

### Official Review · Reviewer_3zff · 2026-02-15
**Clear Relevance and Contribution**

**Rating:** 7
**Confidence:** 3

**Review:**

## Review

### Relevance to Workshop Topics
This paper is most relevant to the **Geometry in theoretical analysis** track, particularly under **Data and latent geometry**. The work provides insight into the geometric structure of latent representations in large language models, showing that hyperbolic probe geometries better preserve reasoning information than Euclidean probes. The analysis of layer-wise degradation and representational compression aligns with the workshop’s focus on understanding latent geometry through geometric tools.

---

### Originality and Novelty
The paper offers a novel perspective by comparing Euclidean and hyperbolic probing geometries in reasoning-specialized versus standard instruction-tuned LLMs.

---

### Technical Soundness
The methodology appears technically sound and well-motivated. Overall, the results are convincing. Limitations remain, such as 4-bit quantization and limited experimental coverage, as stated explicitly in the paper.

---

### Clarity and Organization
The paper is clearly written and well-organized.

---

### Overall Assessment
This work makes a valuable contribution to understanding how non-Euclidean geometry, particularly hyperbolic space, can provide robustness advantages in probing reasoning representations in LLMs. It fits well within the workshop’s goals and raises interesting directions for future research on geometric structure in large-scale models.

**Pmlr Suitability:**

NA

---

### Official Review · Reviewer_n8tE · 2026-02-22
**Interesting Empirical Finding on Hyperbolic Probe Robustness, but Limited Scope and Insufficient Background**

**Rating:** 5
**Confidence:** 3

**Review:**

Summary

This paper investigates whether hyperbolic geometry provides advantages over Euclidean geometry for probing reasoning structure in LLM hidden states. The authors compare Euclidean and hyperbolic distance-based probes on PrOntoQA, evaluating a reasoning-specialized model (DeepSeek-R1-Distill-Qwen-7B) and a standard instruction-tuned model (Qwen2.5-7B-Instruct).
The key finding is that at the final layer of the reasoning-specialized model, Euclidean probes degrade substantially (Spearman ρ ≈ 0.49), whereas hyperbolic probes remain robust (ρ ≈ 0.97). This degradation is not observed in the standard instruction-tuned model. The authors attribute this effect to severe representational compression at the final layer and argue that hyperbolic geometry is more robust under such conditions.

Strengths
- The paper identifies a clear and somewhat surprising empirical phenomenon: final-layer Euclidean probe degradation in a reasoning-specialized model, contrasted with stable hyperbolic performance.
The layer-wise analysis and activation statistics (e.g., participation ratio and isotropy) provide a plausible explanation for the observed effect.
- The experimental setup is systematic and includes useful ablations over dimension and curvature.
For a Tiny Paper submission, the scope is appropriate and focused.

Weaknesses
- The probed structure is very simple (reasoning depth 1–5), and the ground-truth metric is 1D and Euclidean-embeddable. Thus, the results do not establish that hyperbolic geometry is theoretically necessary, only that it appears more robust in this particular setup.
- The evaluation is limited to a single synthetic dataset and two 7B models sharing the same backbone, which limits generality.
- The Euclidean baseline may be sensitive to conditioning effects under final-layer compression; additional controls (e.g., whitening or rescaling) would strengthen the geometric interpretation.

Writing / Presentation

I found the write-up difficult to follow. The paper assumes substantial background knowledge in hyperbolic geometry and probing but provides very little introductory explanation. Key concepts (e.g., why hyperbolic space is suitable for hierarchies, what structural probes do, why depth differences are used as distances) are not clearly motivated for a broader audience. As written, the paper feels more like a condensed technical note than a self-contained Tiny Paper. Improving clarity and providing minimal conceptual background would significantly strengthen accessibility and impact.

Overall Assessment

Despite its limitations, the paper presents an interesting empirical observation about geometry-dependent probe behavior in reasoning-specialized models. While the claims about “hyperbolic geometry of reasoning” should be interpreted cautiously, the phenomenon itself is worth sharing with the community.

Recommendation: Weak Accept.

The contribution is narrow but potentially useful, and appropriate for a workshop Tiny Paper, provided the authors improve clarity.

**Pmlr Suitability:**

NA

---

### Official Review · Reviewer_Kguo · 2026-02-24
**Hyperbolic probes resist late-layer degradation in reasoning LLMs, but limited model/dataset diversity**

**Rating:** 6
**Confidence:** 4

**Review:**

This paper probes DeepSeek-R1 and Qwen2.5 on PrOntoQA with Poincare ball vs Euclidean probes, finding hyperbolic probes maintain $\rho \approx 0.97$ across layers while Euclidean probes degrade to $\rho = 0.488$ at the final layer of the reasoning model. Thinking tokens are shown to concentrate hierarchical information at the compressed final layer.

**Pros:**
- Clean cross-model design isolating reasoning fine-tuning effects on the same Qwen2.5 backbone
- Thorough ablations (dimension $d \in \{2,...,32\}$, curvature $c \in \{0.1,...,1.0\}$) and activation statistics
- Thinking token geometric analysis ($\rho=0.871$ vs $0.390$ all-pool) provides geometric validation of Qian et al. (2025)
- Strong reproducibility: fixed seed, 5-fold CV, code repo, detailed hyperparameters

**Cons:**
- Only 2 models sharing the same backbone , cannot distinguish general reasoning specialization from this specific distillation
- The 1D ordinal target ($|depth_i - depth_j| \in \{0,...,4\}$) embeds isometrically in both geometries. A regularized Euclidean baseline or branching tree targets would better isolate the geometric contribution
- Regex-based thinking token identification (6.7% rate vs Qian et al.'s 0.5–5%) , no random baseline of same size
- Single synthetic dataset (PrOntoQA) with no natural language reasoning evaluation

**Pmlr Suitability:**

NA

---

### Meta-Review · Area_Chair_rKiQ · 2026-02-24

**Decision:**

Accept

**Metareview:**

This paper studies probing reasoning in LLM using novel ideas from hyperbolic geometry. It is clean, interesting, relevant, and novel. I recommend acceptance and suggest that the authors incorporate the reviewers' comments into their final version.

**Relevance To Proceedings:**

Tiny paper — does not apply

**Relevance To Workshop:**

Yes — suitable for GRaM

---

### Decision · Program_Chairs · 2026-03-02

Accept (Poster)